# cDNA Cloning of Feline PIWIL1 and Evaluation of Expression in the Testis of the Domestic Cat

**DOI:** 10.3390/ijms24119346

**Published:** 2023-05-26

**Authors:** Leanne Stalker, Alanna G. Backx, Allison K. Tscherner, Stewart J. Russell, Robert A. Foster, Jonathan LaMarre

**Affiliations:** 1Department of Biomedical Sciences, Ontario Veterinary College, University of Guelph, Guelph, ON N1G 2W1, Canada; 2Department of Pathobiology, Ontario Veterinary College, University of Guelph, Guelph, ON N1G 2W12, Canada

**Keywords:** testis, spermatogenenesis, gamete, PIWI pathway, small non-coding RNA, transposable element, male reproductive models

## Abstract

The PIWI clade of Argonaute proteins is essential for spermatogenesis in all species examined to date. This protein family binds specific classes of small non-coding RNAs known as PIWI-interacting RNAs (piRNAs) which together form piRNA-induced silencing complexes (piRISCs) that are recruited to specific RNA targets through sequence complementarity. These complexes facilitate gene silencing through endonuclease activity and guided recruitment of epigenetic silencing factors. PIWI proteins and piRNAs have been found to play multiple roles in the testis including the maintenance of genomic integrity through transposon silencing and facilitating the turnover of coding RNAs during spermatogenesis. In the present study, we report the first characterization of PIWIL1 in the male domestic cat, a mammalian system predicted to express four PIWI family members. Multiple transcript variants of *PIWIL1* were cloned from feline testes cDNA. One isoform shows high homology to *PIWIL1* from other mammals, however, the other has characteristics of a “slicer null” isoform, lacking the domain required for endonuclease activity. Expression of PIWIL1 in the male cat appears limited to the testis and correlates with sexual maturity. RNA-immunoprecipitation revealed that feline PIWIL1 binds small RNAs with an average size of 29 nt. Together, these data suggest that the domestic cat has two PIWIL1 isoforms expressed in the mature testis, at least one of which interacts with piRNAs.

## 1. Introduction

The roles of small non-coding RNAs in spermatogenesis have become subjects of intense investigation over the past decade due to their involvement in many cellular processes including gene regulation and genome defense [1,2,3]. One important family of small non-coding RNAs (sncRNAs) consists of single-stranded RNA molecules 24–32 nucleotides in length, known as piRNAs [4,5]. These small RNAs associate with PIWI proteins from the Argonaute family, forming complexes known as piRNA-induced silencing complexes (piRISCS). piRNAs guide piRISCS to specific target sequences through base-pair complementarity, where they direct downstream post-transcriptional silencing through PIWI endonuclease-mediated slicing activity [6,7,8,9,10,11]. In addition, piRISC complexes can also form nuclear complexes with other proteins to exert control over epigenetically repressive chromatin hallmarks such as histone 3 lysine 9 (H3K9) methylation and DNA methylation, leading to silencing of targets at the transcriptional level [12,13,14,15,16].

A major role of PIWI proteins is the maintenance of genomic integrity through the silencing of transposable elements (TEs) [17,18,19,20,21]. In somatic cells, TEs are normally silenced epigenetically, but during spermatogenesis, epigenetic marks are removed and TEs become actively transcribed, increasing the risk of genome damage through transposition [22]. In developing spermatocytes, piRNAs with complementarity to TEs guide piRISCs to TE transcripts, leading to slicer-mediated TE transcript decay and co-transcriptional silencing [19,21,22,23,24,25,26,27]. Alterations in PIWI expression in many model systems correspond with hypomethylation of TE loci, over-expression of TE transcripts, and failure of meiosis, typically resulting in spermatogenic arrest and sterility [3,7,13,28,29,30,31]. In PIWI-null mice, sterility is observed only in the male [28,29] whereas several elegant recent studies suggest that PIWIL1 suppression in hamsters leads to male and female sterility [30,31]. In the context of model systems for gametogenesis, this is particularly relevant to human biology, as humans and hamsters express four PIWI proteins (PIWIL1, L2, L3, and L4) whereas mice express only three (PIWIL1, L2, and L4).

In all mammals examined to date, PIWIL1 is required for spermatogenesis, with PIWIL1 null spermatocytes arresting at the pachytene stage of meiosis, resulting in azoospermia [27,28,29]. The effect of PIWIL1 suppression on male fertility appears to result from both failure of TE silencing [32] and changes in the turnover and elimination of specific mRNAs during spermatogenesis directed by a class of piRNAs [33]. In the adult mammalian testis, only some piRNAs appear to target TEs, whereas a large subset of the piRNA population target transcripts from somatic genes and lncRNAs expressed at specific stages of spermatogenesis [13,34,35,36,37,38,39,40]. This supports a growing series of observations that PIWIs play roles in addition to TE silencing such as regulating mRNA decay and turnover [33,40,41,42]. Collectively, these studies strongly indicate that the PIWI system plays an active role in gene and TE regulation during spermatogenesis.

Understanding the roles of specific pathways that mediate spermatogenesis in domestic mammals such as the cat is relevant from several perspectives; it provides an accessible model of the process in humans [43], and may facilitate the development of novel approaches to both contraception/overpopulation in feral cats [44] and species preservation in threatened or endangered Felidae [44]. PIWI expression has been examined in mammals such as the dog [45], which has three PIWI family members, the cow [46,47], pig [48,49,50,51], and hamster [30,31], which have four PIWI family members. PIWI proteins have not been cloned or characterized in Felidae. In the present study, we have cloned feline *PIWIL1*, an important member of the PIWI family, and have detailed several aspects of PIWI expression in the cat testis. This study represents the first published investigation of the PIWI pathway in the cat and provides insight into PIWI biology in an additional small mammal model that, like humans, expresses four PIWI family members.

## 2. Results

### 2.1. Identification of Two PIWIL1 Isoforms in the Feline Genome that Differ in Their Coding Regions

To clone feline *PIWIL1*, we first analyzed predicted sequences from publicly available databases such as NCBI and ENSEMBLE. Several transcript variants of feline *PIWIL1* were predicted, with sequence differences present both within the 5′ untranslated region (5′UTR) and within the coding sequence. Further characterization of these variants was performed by comparing predicted sequences to the genomic sequence of the *PIWIL1* gene, the alignment of which is detailed in Figure 1. The complete feline *PIWIL1* gene contains a total of 26 exons which vary markedly in size. 

Transcript variants X1 through X4 (XM_006938362.3, XM_006938363.3, XM_019814415.1, and XM_019814416.1, respectively), were found to have variation within the 5′UTR but identical coding regions. Transcript X5 (XM_006938364.2), however, shows lower sequence homology in the coding region; exons 6 through 22 are present, exon 23 is lacking, and exons 24, 25, and part of exon 26, are present, resulting in a shorter final transcript. To reduce confusion of nomenclature, cDNAs derived from transcript variants X1 through X4 will be referred to as *PIWIL1 Isoform 1* or *PIWIL1-1* (due to their identical coding regions) and transcript variant X5 will be referred to as *PIWIL1 isoform 2* or *PIWIL1-2*.

### 2.2. cDNA Cloning and Evolutionary Conservation of Feline PIWIL1 Isoforms

Differences within the coding regions of the PIWIL1 isoforms prompted us to design primers to amplify the different predicted sequences using alternate reverse primers based on sequences at termini of the different coding regions (exon 23, or 26, respectively). RT-PCR amplification was performed from mature feline testes tissue and resulted in 2689 bp and 2586 bp products for *PIWIL1-1* and *PIWIL1-2,* respectively. Products were then cloned into expression plasmids pcDNA3.1 and sequenced via automated DNA sequencing using overlapping sequencing primers (Appendix A).

Cloned *PIWIL1-2* cDNA showed no amino acid substitutions and 100% alignment with the predicted X5 variant (GenBank Accession Number MG209613). Cloned *PIWIL1-1* however, contained several nucleotide substitutions from the predicted PIWIL1 Isoform X1 sequence (XP-019669974.1) predicted to result in a single amino acid alteration of K:E at position 796. An additional cytosine is predicted between position 2500–2503, resulting in a frame-shift mutation from this amino acid position 835 to the C-terminal end of the protein. Interestingly, this additional nucleotide was observed in several cloning attempts from different individual animals (GenBank Accession number MG209612). 

The sequence differences between the coding regions of *PIWIL1-1* and *PIWIL1-2* prompted us to examine the sequence characteristics of the functional protein domains widely considered to be required for PIWI protein binding and activity, including the presence of both a PAZ and PIWI domain, as well as maintenance of a functional catalytic triad. PAZ domains are required for the binding of piRNAs to PIWI proteins [51,52] and PIWI domains are involved in the regulation of PIWI catalytic activity, with slicer activity conferred by the presence of a DDH catalytic triad within this domain [11,19,53]. *PIWIL1-2* encodes an 861AA protein, containing a complete PAZ domain from amino acid 276–393 and a PIWI domain from amino acid 401–844, representing a full piRNA binding domain, as well as a 5′RNA guide site and active site containing a DDH catalytic triad at positions 632, 702, and 836, respectively. *PIWIL1-1* encodes an 896AA protein that contains a full PAZ domain. However, the PIWI domain from amino acids 401–732 in this variant is truncated, lacking a complete catalytic triad, with a serine replacing the histidine at position 836. This suggests that the PIWIL1-1 variant lacks a complete enzymatic active site due to an incomplete, or mutated, catalytic triad. 

Optimal global alignment of the full-length *Felis catus* PIWIL1-1 and PIWIL1-2 protein sequences to selected mammalian organisms: human (*Homo sapiens*), mouse (*Mus musculus*), dog (*Canis lupus familiaris*), and cow (*Bos taurus*) was then performed using MUSCLE alignment with 8 allowable iterations (see Appendix A). PIWILI-2 was found to have high homology to all tested mammalian organisms (over 95%) (Appendix A). PIWIL1-1 had distinctly less overall homology, ranging from 79.4% to 82.0% resulting from a lack of homology within the PIWI domain (Appendix A). 

### 2.3. Expression of Feline PIWIL1 cDNA Clones

To further characterize feline PIWIL1, we independently transfected pcDNA3.1-PIWIL1-1 (pcDNA-PL1) and pcDNA-PIWIL1-2 (pcDNA-PL2) into a mammalian cell line (HEK293) that lacks endogenous PIWIL1 expression. Western blots employing an anti-PIWIL1 antibody raised to the N-terminus of the protein (ab105393) demonstrate a band at 90 kDa in feline PIWIL1-2 transfected cells, and a band at a slightly higher molecular weight in feline PIWIL1-1 transfected cells (Figure 2a). No band was evident in HEK293 cells transfected with pcDNA3.1 alone, or in cells transfected with pcDNA3.1-PIWIL2 (Figure 2a). Western blots utilizing a PIWIL1 antibody raised to the C-terminus of PIWIL1 (ab12337) show a specific band at 90 kDa in feline PIWIL1-2 transfected cells, but no band is evident in feline PIWIL1-1 transfected, pcDNA3.1 alone transfected, or feline PIWIL2 transfected HEK293 (Figure 2b). These data reveal that exogenous PIWIL1 is expressed in transfected cells and confirms the recognition of the PIWIL1-2 variants in the cat by both antibodies utilized and the absence of cross-reactivity with another PIWI family member (PIWIL2). We were unable to identify or develop any antibody specific for the feline PIWIL1-1 isoform despite multiple attempts. 

### 2.4. PIWIL1 Protein Expression Is Restricted to the Testis in the Male Cat

The expression of both PIWIL1 isoforms was next evaluated in a panel of male feline tissues, to observe protein expression across the entire organism. Organs, including heart, liver, lung, spleen, kidney, brain, and small/large intestine, were compared to testes, where PIWIL1 expression is expected. Unfortunately, definitive discrimination of the two feline PIWIL1 isoforms is not possible with available antibodies. To evaluate the isoforms to the best of our ability with available reagents, Western blotting was first performed using the C-terminal PIWIL1 antibody that recognizes the PIWIL1-2 variant only. PIWIL1-2 protein expression is limited to testes, with no protein expression evident in any other organs tested (Figure 3a). This was followed with an evaluation of the same tissue samples using an N-terminal PIWIL1 antibody that recognizes both PIWIL1-1 and PIWIL1-2 variants (Figure 3b). PIWIL1 expression was also limited to the testis using this antibody. Overall, PIWIL1 appears to be expressed exclusively within the testis in the cats we evaluated, which is consistent with previously described studies in other mammalian species.

### 2.5. Expression and Localization of Feline PIWIL1 Changes with Sexual Maturity

PIWIL1 expression was next evaluated in the testis tissue of cats at different levels of sexual maturity. Maturity was determined by the presence or absence of elongated spermatids and overall morphology in parallel-processed hematoxylin and eosin (H&E) stained slides. Standard endpoint RT-PCR was performed using primers for each full-length isoform, revealing that both *PIWILI-1* and *PIWIL1-2* were expressed in 3/3 mature cats tested, but were absent from immature animals under the PCR conditions employed (Figure 4a). 

Quantitative real-time PCR (qRT-PCR) was then used to evaluate this expression pattern with greater sensitivity in individual animals. Due to the limited sample quantity, different animals of similar age were used for the qRT-PCR analysis than for the standard RT-PCR. The presence of exon 23 in *PIWIL1-1* which is absent in *PIWIL1-2* (see Figure 1) allowed us to design qRT-PCR primers specific for each *PIWIL1* isoform. As in previous studies with other species [45], animal-to-animal variability was observed at the transcript level for *PIWIL1* expression, with relative expression levels observed to be consistently highest in all three mature animals for both isoforms. (Figure 4b,c). When individuals were grouped by maturity, mean transcript expression levels were found to be significantly higher in mature animals for both *PIWIL1-1* and *PIWIL1-2* (*p* = 0.002 and 0.004, respectively) (Figure 4b,c). 

Transcript expression analysis was followed by an evaluation of PIWIL1 protein expression using Western blotting. Samples were first evaluated using the PIWIL1 C-terminal antibody, for the recognition of PIWIL1-2. Western blots demonstrated detectable PIWIL1-2 protein expression in 3/3 mature feline testes samples with no observable expression in 3/3 immature feline testes counterparts (Figure 5a). The samples were then subjected to Western blotting using the N-terminal PIWIL1 antibody that recognizes both PIWIL1 (PIWIL1-1 and PIWIL1-2) isoforms and a similar expression pattern was observed (Figure 5b). Consistent with the initial findings in the tissue panel study, only a single band around 90 kDa was evident upon blotting with the N-terminal antibody. 

We next employed immunohistochemistry to evaluate PIWIL1 cellular localization within the mature and immature feline testes. Three immature and three mature samples were stained using the C-terminal PIWIL1 antibody. Representative images are presented in Figure 5. No staining was observed in the seminiferous tubules of pre-pubertal kittens (Figure 5c). Staining was consistently observed in the cytoplasm of round spermatids, and in elongated spermatids, with the exception of cells that had little detectable cytoplasm in mature cats (Figure 5d). No nuclear or membrane staining was apparent. Lower levels of staining were detected in the interstitial endocrine cells of cats of all ages (2 months old: Figure 5c; 1 year old: Figure 5d). In post-pubertal male cats, staining was seen in pachytene spermatocytes but not spermatogonia, leptotene spermatocytes, or zygotene spermatocytes. Overall, these data clearly demonstrate that feline PIWIL1 transcript and protein expression shows cytoplasmic localization in the testis of sexually mature cats with active spermatogenesis, supporting established roles in spermatogenesis for PIWIL1 in this species.

### 2.6. Feline PIWIL1 Binds Small RNA with the Predicted Size of piRNAs

PIWI protein binding to small RNAs called piRNA that average 29 nt in length is a defining feature of this family that is essential for their function. To determine whether feline PIWIL1 formed complexes with piRNAs in feline testis, RNA immunoprecipitation, and gel electrophoresis were employed. Briefly, PIWIL1 was immunoprecipitated from mature feline testes tissue using the N-terminal PIWIL1 antibody. Each immunoprecipitation was performed in duplicate, with one replicate being used for protein analysis, and one used for the evaluation of bound RNAs. Western blotting to verify the specificity of the pull-down was performed using the C-terminal PIWIL1 antibody, which demonstrated the presence of a specific band at the expected molecular weight in both the input and immunoprecipitate lanes, indicating specific enrichment of PIWIL1 (Figure 6a). The immunoprecipitates utilized for RNA analysis were subjected to RNA extraction to isolate associated RNAs, after which the bound RNA population was visualized using gel electrophoresis (TBE, Urea) and SYBR-Gold staining. The molecular size of the stained RNA population in the PIWIL1-IP samples was determined relative to the positions of the RNA molecular weight markers (using a logRF calculation), revealing a small RNA population 29 nt in length in PIWIL1 IP samples (Figure 5b). 

Identical samples were subjected to the precipitation process using either beads alone, or non-specific IgG, and showed no associated PIWIL1 protein or bound small RNA populations. These data reveal that feline PIWIL1 binds a distinct small RNA population that is consistent with the established size range of piRNAs that interact with PIWIL1 in other species and strongly suggests that the protein is active in mature feline testes.

## 3. Discussion

The studies presented here provide the first identification of a PIWI protein family member in the cat and report the pattern of PIWIL1 expression in the mature feline testis. We demonstrate that multiple transcript isoforms of PIWIL1 are present in this species, at least one of which encodes a protein that binds small RNAs in a size range consistent with piRNAs. PIWI family members have been found to play important roles in the maintenance of spermatogenesis through the silencing of retrotransposons and through promoting the decay of mRNAs [3,7,13,31]. In particular, PIWIL1 is the major PIWIL1 family member responsible for binding pachytene piRNAs, a subclass of piRNAs that comprise approximately 80% of the small RNAs present in mature mouse testes, and pi6, a conserved piRNA responsible for spermatogenesis [53]. Based on this and other studies demonstrating its functional importance in spermatogenesis, we initiated our investigation of the PIWI pathway in the feline testis by cloning and characterizing PIWIL1. Complementary DNAs encoding multiple isoforms of *Felis catus PIWIL1* were cloned and sequenced, and PIWIL1-1 and PIWIL1-2 were exogenously expressed in HEK cells to confirm protein identity. In vivo expression levels were examined, and the expression of both isoforms is correlated with sexual maturity, as determined by transcript levels in the testis. PIWIL1-2 was also correlated with sexual maturity as determined by protein expression and showed cytoplasmic localization by immunohistochemistry. *Felis catus* PIWIL1 was also found to associate with small RNAs in the predicted size range of piRNAs, suggesting that at least one PIWIL1 isoform is functional in the cat.

Using the NCBI database, our initial in silico examination of PIWIL1 in the cat predicted multiple transcript variants. Although such variants are common in mammals, most of these show sequence differences outside of protein-coding regions, either within the 5′ or 3′ UTRs. For the five predicted PIWIL1 transcript variants in the cat, only two appear to have different coding regions. We, therefore, focused our attention on these. *PIWIL1-2* showed 100% alignment with the predicted sequence and showed high homology with previously characterized PIWIL1 transcripts from other mammalian species. Interestingly, the *PIWIL1-1* variant cloned did not show 100% alignment with the predicted sequence, with an additional cytosine located in the middle of the C-terminal region, leading to a frame-shift mutation affecting the C-terminal end of the encoded protein. This extra cytosine was present in multiple transcripts sequenced from multiple animals, suggesting that it represents a true sequence variant as opposed to a PCR-generated mutation or sporadic sequencing error. To examine this variant further, we accessed the NCBI SNP database and compared our cloned sequence to other known SNP variants. Although SNP data in the feline is limited, we found that an additional cytosine at positions 2500–2503 (in a run of four cytosines) was observed in 4/5 sequenced SNPs from this region, suggesting that this sequence more accurately represents the wild-type feline *PIWIL1-1* than the predicted sequence provided at NCBI. 

The predicted PIWIL1-1 protein sequence shows low homology with both PIWIL1-2 and PIWIL1 from other mammalian species. Although the N-terminal region of the protein, including the piRNA binding PAZ domain is similar, substantial divergence is present within the PIWI domain. Major differences in the C-terminus of the protein are predicted to alter the structure of the PIWI domain, including the elimination of a functional catalytic DDH triad, through the substitution of histidine (H) with serine (S) at position 836. The DDH catalytic triad is similar to the DDE catalytic triad of RNase H and is required for the enzymatic “slicer” activity of PIWI proteins [52,54,55]. This suggests that the feline PIWIL1-1 variant may not possess endonuclease activity. Genetic differences within this region have been found to result in male sterility and increased retrotransposon expression in spermatogonial cells [6,32].

To determine the extent to which the PIWIL1-1 variant is evolutionarily conserved in *Felidae*, we investigated several species with common ancestry to the domestic cat. Although full genome sequence information was not available for any other members of the genus *Felis* at the time of the comparison, predicted transcripts for PIWIL1 exist for several family members. Genomic information is available for *Panthera tigris* (tiger) and *Panthera pardus* (leopard), which share a common ancestor to the domestic cat, diverging approximately 10.8 million years ago, and *Acinonyx jubatus* (cheetah), which shared a common ancestor to the domestic cat roughly 6.7 million years ago [56]. Computational analysis predicted five transcript variants of PIWIL1 in *P. pardus* and one PIWIL1 transcript in *P. tigris* (which aligns with 100% coverage and 99% identity to *P. pardus* variants 1–3). *P. tigris* and all five variants of *P. pardus* show strong conservation with *F. catus* PIWIL1-2, and strong conservation to the N-terminus of PIWIL1-1, although they deviate from PIWIL1-1 at exon 23. Unlike the genus *Panthera,* the genetically closer relative *A. jubatus* has two PIWIL1 variants that are homologous to the PIWIL1-1 and PIWIL1-2 variants in the present study. Furthermore, this species demonstrates the only other predicted example of a PIWI-domain truncated variant of the PIWIL1 protein. Interestingly, *A. jubatus* PIWIL1-1 also contains a fourth cytosine at position 2500, further supporting our contention that this is a more representative sequence in these species. In comparison with other mammalian species, based on the prediction that it is functionally intact, PIWIL1-2 is likely to be a more highly conserved transcript. The truncated PIWI-domain isoform PIWIL1-1 may be relatively unique to *Felidae* and conserved among the closest relatives of the domestic cat, reflecting a more recently acquired genetic alteration. 

In the current study, feline PIWIL1-1 and -2 were expressed successfully in HEK293 cells. To determine the specificity of both PIWIL1 antibodies, and to confirm the identity of the PIWIL1-1 and PIWIL1-2 clones, we performed a Western blot analysis of testicular lysates. This analysis revealed an expected single band in both PIWIL1-1 and PIWIL1-2 transfected lanes, with the larger PIWIL1-1 isoform migrating at a slightly higher molecular weight. Both PIWIL1 antibodies were specific for PIWIL1, showing no signal in lysates from PIWIL2 transfected cells, or control lanes. Both PIWIL1 variants were recognized by the antibody directed at the N-terminus, whereas only the PIWIL1-2 variant was recognized by an antibody directed at the PIWIL1 C-terminus. Interestingly, only a single band was observed in the feline testes control lane using either antibody, suggesting that there may be only one PIWIL1 protein variant expressed in vivo. However, it does appear that the endogenous PIWIL1 from testes migrates at a slightly lower molecular weight than either exogenously expressed PIWIL1 isoform, suggesting that post-translational modifications affecting electrophoretic migration patterns may be present in the cloned constructs or endogenous protein. Further individual characterization will require the development of isoform-specific antibodies.

In many other mammalian species, PIWIL1 expression is highly restricted to the testes in both developing and mature animals [27,45,46,57]. The pattern of PIWIL1 expression reported here is consistent with that reported in other species using either an N-terminal or C-terminal-specific antibody. This suggests that if multiple protein isoforms are present, they share a tissue-restricted expression pattern. Our evaluation of PIWIL1 expression and binding to piRNAs in the context of the whole testis focused on PIWIL1-2 due to antibody specificity limitations. Results demonstrated strong expression of PIWIL1 in the cytoplasm of pachytene spermatocytes, round spermatids, and elongating spermatids, showing the presence of clear cytoplasm in mature animals. Interestingly, staining was completely absent from the testes of immature animals, which differs from our canine studies, where staining was also observed in prespermatogonia and spermatogonia in immature dogs [45]. Other differences between the present feline work and our previous canine studies include the expression of multiple PIWIL1 isoform transcripts. Transcript expression of both full-length PIWIL1 isoforms is absent when assessed by standard RT-PCR, although quantitative RT-PCR revealed extremely low expression of both isoforms in the immature animals tested. Immature feline samples lack any detectable expression of PIWIL1 protein of either isoform, supporting our IHC results. Based on its known functions, we postulated that PIWIL1 expression would be regulated over sexual maturation, with mature animals demonstrating higher levels of PIWIL1 expression. This is consistent with the predicted role of PIWIL1 in the meiotic stages of spermatogenesis as mature animals have a higher number of developing germ cells that would require PIWIL1 expression for the regulation of many piRNA targets, particularly pachytene spermatocytes. The differences in the timing of PIWIL1 expression in the cat versus the mouse suggest that some functional differences may exist between the two species. Future studies to determine the precise developmental stage at which PIWIL1 expression begins in the cat, the stimuli that drive this expression, and the roles it may play relative to other mammals that show slightly different patterns are clearly warranted. 

The studies presented here suggest that feline PIWIL1 is functionally similar to the protein in other mammals. RNA-IP revealed that PIWIL1 binds short RNA species within the expected size range of piRNAs (29 nt), with no obvious shorter RNAs from other small RNA classes. Although the requirement for slicer activity of all PIWI family proteins has been debated, all functional studies with PIWIL1 reveal the requirement of piRNA binding for function [33,58]. The data presented here, therefore, suggests that at least one PIWIL1 protein is functional in the feline testis.

The presence of transcript variants is one particularly interesting observation in the present study. Many mammals have multiple PIWI transcripts, although most remain predicted rather than confirmed at the mRNA or protein levels. Notably, humans have two verified isoforms of PIWIL1 (NM_004764.4 and NM_001190971.1); one that aligns closely with other species (including the feline) (NM_004764.4) and a second, shorter isoform (NM_001190971.1). This shorter isoform differs substantially in length from the feline *PIWIL1-1* identified here but is similar with respect to the presence of a complete N-terminal region and PAZ binding domain while lacking the third element of the DDH catalytic triad required for endonuclease activity. Previous work in our laboratory has also characterized multiple PIWIL1 isoforms named *PL1-10* (KP770145) and *PL1-13* (KP770146) in the bovine [47]. These alternatively spliced *PIWIL1* isoforms were identified in the bovine testes and in bovine oocytes, and maintain N-terminal regions required for Tudor domain protein binding and piRNA complex formation, however, they lack a PIWI domain, and are likely deficient in target recognition/slicing activity [52]. By analogy with other unrelated pathways, potential roles for PIWIL1 isoforms that lack one or more functional domains could include activity as dominant negative pathway inhibitors or as piRNA or mRNA “traps” [59]. Ultimately, it will be important to determine with functional studies the specific roles that catalytically inactive variants of PIWIL1 may play in the control of transposable elements, gene regulation, and male fertility.

## 4. Materials and Methods

### 4.1. Antibodies

All antibodies utilized within this body of work are listed in the antibody verification table (Appendix A), including supplier information, catalog number, RRID number, and use. All PIWIL1 antibodies were validated for use in the feline as detailed within the manuscript. All antibodies utilized have been published previously for the same application in different species [45,46].

### 4.2. Cell Culture 

HEK293 cells were cultured in high glucose DMEM (Thermo Fisher, Mississauga, ON, Canada) containing 10% fetal bovine serum (FBS), 4 mM L-glutamine, and 1% penicillin–streptomycin antibiotic/anti-mycotic (Thermo Fisher, Mississauga, ON, Canada) at 37 °C/5%CO_2_. 

### 4.3. RNA Isolation and cDNA Synthesis

Total RNA was isolated from 8 feline testes samples (~500 mg tissue) individually using the Qiagen RNeasy Mini Kit (Qiagen, Toronto, ON, Canada) and treated with DNAse 1 (Qiagen, Toronto, ON, Canada) during the isolation protocol according to manufacturer’s instructions. RNA concentrations were determined using a Nanodrop spectrophotometer (Thermo Fisher, Mississauga, ON, Canada) and, for cloning reactions, 1 µg total RNA was reverse transcribed using oligo(dT)12–18 primer and Superscript III™ reverse transcriptase (Thermo Fisher, Mississauga, ON, Canada). RNA (1 µg), oligo(dT) primer (1 μg), and 1 µL 10 mM dNTP mix were combined in a 13 µL reaction volume, incubated at 65 °C for 5 min, and placed on ice for 2 min. Reverse transcription was performed with 200 U/µL of Superscript III™ reverse transcriptase, 0.1 M DTT, 1x buffer, and 40 U/µL RNAse OUT RNAse inhibitor (Thermo Fisher, Mississauga, ON, Canada) in a final reaction volume of 20 µL at 50 °C for 50 min. The reaction was terminated at 70 °C for 15 min. RNAse H (2U: Thermo Fisher, Mississauga, ON, Canada) was added to each sample and incubated at 37 °C for 20 min. Reactions were stored at −20 °C until use. RNA used for RT-PCR analysis was reverse transcribed using Quanta qScript cDNA Supermix (Thermo Fisher, Mississauga, ON, Canada). Briefly, 1 µg total RNA was added to 4 µL of Supermix to a final concentration of 1x in a total reaction volume of 20 µL. The reaction was incubated at 25 °C for 5 min, 42 °C for 30 min, and 85 °C for 5 min as per the supplier’s protocol. Samples were stored at −20 °C until use.

### 4.4. cDNA Cloning of Feline PIWIL1 Isoform 1/PIWIL1 Isoform2/PIWIL2

The feline PIWIL1 coding region was amplified from feline testicular cDNA using Q5 Polymerase High Fidelity (New England BioLabs, Whitby, ON, Canada). Primers for amplification are shown in Appendix A. All PIWI isoforms were amplified from mature domestic shorthair feline testes samples. PCR conditions were as follows: 98 °C for 30 s followed by 5 cycles at 98 °C for 10 s, 52 °C for 30 s, and 72 °C for 100 s. This was followed by 35 cycles at 98 °C for 10 s, 65 °C for 30 s, and 72 °C for 100 s. PCR products were verified for size by electrophoresis on 0.8% agarose/TAE gels, and products were purified using the PureLink Quick Gel Extraction Kit (Thermo Fisher, Mississauga, ON, Canada) as per the manufacturer’s protocol. Forward and reverse primers incorporated restriction digest sites as shown in Appendix A. PCR products and the pcDNA3.1 vector (Thermo Fisher, Mississauga, ON, Canada) were digested for 20 min at 37 °C. An amount of 1 µL calf intestinal alkaline phosphatase (CIAP, Thermo Fisher, Mississauga, ON, Canada) was added to each vector digestion and incubated at 50 °C for 20 min. Digestion products were then PCR purified using the PureLink Quick PCR purification kit (Thermo Fisher, Mississauga, ON, Canada) as per manufacturer’s instructions and the final elution was completed in 20 µL. Products were ligated for 1 h using T4 DNA Ligase (Thermo Fisher, Mississauga, ON, Canada) as per vendor protocol. The products of the ligation reactions were used to transform competent DH5α *E. coli* as per the supplier’s instructions. Positive colonies were isolated from carbenicillin-agar plates and grown for plasmid isolation using a Qiagen Miniprep kit (Thermo Fisher, Mississauga, ON, Canada) according to vendor instructions. Constructs were sequenced at each step to verify orientation, reading frame, and sequence specificity by comparison with predicted sequences in the NCBI database. 

### 4.5. Gene Expression Analysis

Reverse transcription quantitative PCR (RT-qPCR) was employed to analyze the expression level of *PIWIL1-isoform1* and *PIWIL1-isoform 2* in immature (3–4 month) and mature (>9 month) feline testicular samples (see complete list of age and breed information in Appendix A). Briefly, cDNA from qScript reverse transcription was diluted to 1.5 ng/ul. The PCR reaction mix included 5 µL perfeCTa SYBR Supermix (Thermo Fisher, Mississauga, ON, Canada), 0.5 µM primer mix and 3 ng cDNA mixed to a total volume of 10 µL. Reactions were performed in a CFX qPCR (BioRad, Mississauga, ON, Canada) using a 96-well format. No template controls were completed for each primer set. All samples were tested in triplicate. The reaction conditions were as follows: 1 cycle at 95 °C for 2 min, followed by 35 cycles at 95 °C for 10 s and 64 °C for 30 s. Each run was followed by a melting curve analysis to control for primer dimer formation and to ensure accurate amplification. Standard curves were completed to calculate primer efficiency in all cases. Relative transcript level was calculated using the ΔΔCq method using RPS5, SDHa-1, and GAPDH as reference genes in all samples. Relative expression was calculated for the average of all samples. Calculations were performed using Qbase + (BioGazelle) and graphed (PRISM software version 6.0; GraphPad, San Diego, CA, USA). Presented data are log_2_ transformed. Statistical analysis (an unpaired, two-tailed Mann–Whitney test) was performed using GraphPad Prism 6 software, and comparisons with a *p*-value of < 0.05 were considered significant.

### 4.6. Western Blotting

Feline tissues (500 mg) were homogenized using a motorized homogenizer in 300 µL ice-cold lysis buffer (25 mM Tris HCl pH7.4, 150 mM KCl, 2 mM EDTA, 0.5% NP40, 1 mM NaF, 1 mM DTT) with protease/phosphatase inhibitor cocktail (Biotool) until samples were homogeneous. HEK293 cells were washed 2x in PBS and trypsinized. Samples were pelleted at 1000 × G for 5 min, washed 1x in PBS, and re-pelleted at 1000 × G for 5 min. Cell pellets were resuspended in lysis buffer (as above). Tissue and cellular samples were then incubated for 10 min on ice and cleared by centrifugation at 10,000× *g* for 20 min at 4 °C after which cleared supernatant was removed. When required, a second centrifugation step of 10,000× *g* for 10 min at 4 °C was completed to further clarify. Protein concentration was evaluated using the *Dc* Protein Assay Kit (BioRad, Mississauga, ON, Canada). Sample (20 µg) was loaded onto 8% SDS PAGE gels after boiling in 1x SDS loading buffer containing DTT at 95° C for 5 min. PAGE was completed using the mini-PROTEAN gel system and transferred using a wet transfer apparatus (BioRad, Mississauga, ON, Canada). PVDF Membranes (Immobilon-P, Sigma, Oakville, ON, Canada) were incubated for 1–2 h in 5% non-fat milk in TBS with 0.1%Tween 20 (TBS-T) and incubated with the primary antibody of interest in 5% BSA/TBS-T overnight at 4 °C with gentle agitation. Primary antibodies were as follows: Anti PIWIL1 C-terminal (1:1000, ab12337; Abcam Inc., Toronto, ON, Canada), Anti PIWIL1 N-terminal (1:1000, ab105393; Abcam Inc.) (Anti β-Actin (1:1000, #4967; Cell Signaling, Danvers, MA, USA). After washing, secondary antibody was added to membranes in TBS-T/5% non-fat milk for 45 min at room temperature with gentle agitation. Secondary antibody was as follows: anti-rabbit IgG HRP linked secondary (1:5000, #7074S, Cell Signaling Technology), Blots were visualized using Clarity ECL (BioRad, Mississauga, ON, Canada) and a XRS auto imager (BioRad).

### 4.7. RNA Immunoprecipitation (RIP)

Immunoprecipitation: Mature feline testis tissue (approx. 0.250 g) was homogenized in 500 µL lysis buffer (as above) and cell lysate was cleared at 11,700× *g* for 20 min at 4 °C. Lysate was precleared with 50 µL protein-A magnetic beads (BioRad, Mississauga, ON, Canada) for 2 h at 4 °C. An amount of 100 µL of precleared lysate was added to 40 µL protein-A magnetic beads bound to 3 µg PIWIL1 antibody (Ab105393, Abcam Inc.), 2 µg non-specific rabbit IgG (#12-370, Millipore) or control without antibody, respectively, and incubated with rotation for 2 h at 4 °C. Each IP was performed in duplicate. Samples were then washed 4x in 250 µL lysis buffer containing 1 µL/100 µL RNAse OUT (Thermo Fisher) and 1x complete protease inhibitor (Roche, Laval, QC, Canada). Samples were then washed 1x in 250 µL PBS-T (137 mM NaCl, 2.7 mM KCl, 10 mM Na_2_HP0_4_, 1.8 mM KH_2_PO_4_, 0.1%Tween 20) and placed in clean microcentrifuge tubes. Samples were washed 1 final time in 250 µL PBS-T prior to elution. 

Protein elution: 100 µL protein elution buffer (0.2% (*w*/*v*) SDS, 0.1% (*v*/*v*) Tween 20, 50 mM TrisHCL pH 8.0) was added to beads and incubated for 10 min at room temperature with rotation. Beads were magnetized and the elution removed. This elution was then repeated with a second 100 µL and added to original elution to a total elution 1 (E1) volume of 200 µL. An amount of 1 mL ice cold acetone (Sigma, Oakville, ON) was added to E1, mixed thoroughly and incubated at −20 °C overnight. Samples were then spun at 10,200 × G for 15 min at 4 °C to pellet the precipitate. The supernatant was removed, and the pellet was allowed to air dry for 5 min at room temperature. The pellet was then re-suspended in 1x SDS loading buffer and incubated at 95 °C for 10 min.

RNA elution: 50 µL RNA elution buffer (50 mM Tris pH 8.0, 100 mM NaCl, 10 mM EDTA, 1% SDS) was added to the beads and incubated for 10 min at 65 °C with inversion every 2 min to mix. The magnet was applied, the supernatant was removed and labeled “RNA Elution”. This elution was repeated with an additional 50 µL of RNA elution buffer and added to the RNA elution for a total elution volume of 100 µL. 

RNA extraction: 10 volumes of Qiazol RNA extraction reagent (Qiagen) was added to RNA Elution and RNA was extracted using an RNeasy Micro RNA extraction kit (Qiagen) using the manufacturer’s instructions for enriching small RNA. All samples were eluted in 14 µL RNase free water.

TBE-UREA PAGE: Briefly, 2x TBE UREA denaturing buffer (BioRad, Cat #161-0768) was added to RIP samples to a final concentration of 1x and incubated at 70 °C for 10 min. Electrophoresis was performed using a 15% TBE UREA polyacrylamide gel (BioRad, Cat# 456–6053) in 0.5X TBE buffer (BioRad, Mississauga, ON, Canada) at 200 V for 45 min. An amount of 1 µg total RNA was loaded as a running control. Visualization was achieved after staining with 1x SYBR GOLD (Invitrogen) in 1X TBE buffer for 45 min on an XRS imager using the amber filter. Band size was quantified using ImageLab version 5.0 software (BioRad, Mississauga, ON, Canada).

### 4.8. Immunohistochemistry 

Mature and immature feline testes samples were fixed in 10% neutral buffered formalin overnight at room temperature, dehydrated in isopropanol and embedded in paraffin. Serial 5 µm sections were obtained and mounted on charged glass slides. Slides were deparaffinized and rehydrated in xylene and isopropanol. Endogenous peroxidase activity was blocked with 3% hydrogen peroxide (Sigma, Oakville, ON) for 10 min. Slides were washed in PBS (137 mM NaCl, 2.7 mM KCl, 10 mM Na_2_HP0_4_, 1.8 mM KH_2_PO_4_) followed by citrate buffer antigen retrieval (10 mM citric acid with 0.05% Tween at pH 6.0) for 12 min at 95 °C. Slides were allowed to cool to room temperature for 20 min and washed with PBS. A wax ring was applied around tissue sections and treated with protein block serum-free (X0909, DAKO Canada, Mississauga, ON, Canada) according to the manufacturer’s instructions for 10 min. Slides were incubated with primary rabbit anti-PIWIL 1 antibody (1:100, Ab12337, Abcam Inc.) in PBS with adjacent slides incubated in PBS as negative controls. Slides were incubated in humidified chamber at 4 °C overnight. After incubation, slides were washed in PBS and then incubated with biotinylated secondary anti-rabbit (1:100, B7389, Sigma) for 2 h at room temperature. Slides were then washed with PBS and tertiary ExtrAvadin^®^-Peroxidase (Sigma, Oakville, ON, E2886) was applied at a 1/50 dilution for one hour at room temperature. Slides were then washed with PBS and 3,3′-Diaminobenzidine (DAB) (Sigma, D4293) was added for 30 s after which slides were quickly washed in PBS. Slides were then counterstained in Carazzi’s hemotoxylin for 30 s before sequential water washes and dehydration in isopropanol and xylene. Slides were mounted with Cytoseal (Thermo Fisher, 831016). 

## 5. Conclusions

In conclusion, we have cloned cDNAs encoding PIWIL1 transcript variants that present in the testes of the mature domestic cat and have demonstrated that they share many characteristics of PIWIL1 homologues in other species. The existence of multiple transcript variants is similar to that observed in humans and may facilitate future work on the basic features of this pathway in higher mammals. The physical association of PIWIL1 with small RNAs conforming to the known size of piRNAs in mature cat testis strongly supports a role in RNA turnover during spermatogenesis. Knowledge of these specific variants in Felidae should inform studies aimed at further characterizing the functional roles and regulation of the PIWI pathway during gametogenesis in other mammalian species. 

## Figures and Tables

**Figure 1 ijms-24-09346-f001:**
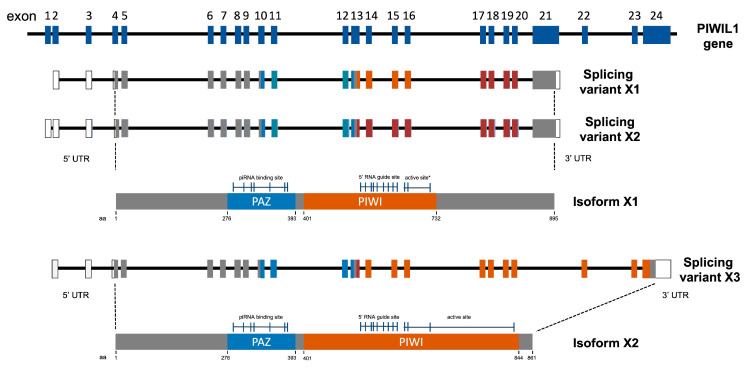
The feline genome encodes multiple transcript variants of PIWIL1. Diagram of encoded transcript variants from the feline genome denoting included and excluded exons in each case. Resulting protein isoforms, PIWIL1-isoform 1 and PIWIL1-isoform 2, respectively, are labeled to indicate important protein domains.

**Figure 2 ijms-24-09346-f002:**
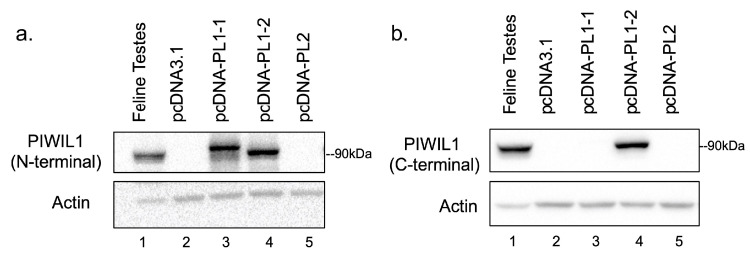
Expression of cloned feline PIWIL1 isoforms. (**a**) Western blot of feline PIWIL1 after transfection with pcDNA-feline PIWIL1-isoform 1 (pcDNAPL1-1), pcDNA-feline PIWIL1-isoform 2 (pcDNAPL1-2), pcDNA-feline PIWIL2 (pcDNAPL2), or pcDNA-3.1 alone into HEK293 cells for 24 h using a PIWIL1 antibody specific to the PIWIL1 N-terminus. Transfected cells compared to mature feline testes, with β-actin as a protein loading control. (**b**) Western blot of feline PIWIL1 after transfection with pcDNA-feline PIWIL1-isoform 1 (pcDNAPL1-1), pcDNA-feline PIWIL1-isoform 2 (pcDNAPL1-2), pcDNA-feline PIWIL2 (pcDNA-PL2), or pcDNA-3.1 alone into HEK293 cells for 24 h using a PIWIL1 antibody specific to the PIWIL1 C-terminus. Transfected cells compared to mature feline testes, with β-actin as a protein loading control.

**Figure 3 ijms-24-09346-f003:**
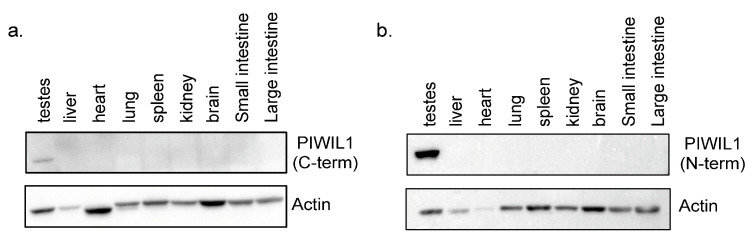
PIWIL1 expression in feline tissues (**a**) Western blot analysis of endogenous PIWIL1 expression in feline tissues using C-terminal PIWIL1 antibody. Actin was used as a protein loading control. (**b**) Western blot analysis of endogenous PIWIL1 expression in feline tissues using N-terminal PIWIL1 antibody. Tissue samples used were identical to tissues in panel (**a**). Actin was used as a loading control. Representative blots shown.

**Figure 4 ijms-24-09346-f004:**
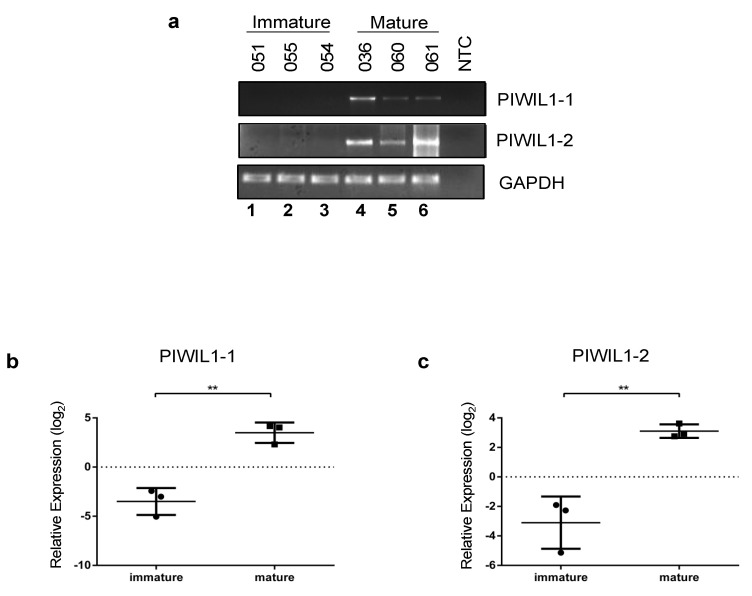
Expression of PIWIL1 transcripts is associated with level of maturity in male cats. (**a**) Endogenous *PIWIL1* expression assessed by standard RT-PCR in cDNA from mature and immature feline testes. PIWIL1isoform 1 and PIWIL1-isoform 2 transcript expression was evaluated using GAPDH as a PCR loading control. (**b**) Relative expression of PIWIL1-isoform 1 and (**c**) PIWIL1 isoform 2 transcripts assessed using quantitative real-time RT-PCR in individual mature and immature feline testis samples. All samples are normalized to SDHa, RPS5, and GAPDH. Log_2_ relative expression is shown with the control (0 value) set to the mean of all samples. ** *p* value < 0.01.

**Figure 5 ijms-24-09346-f005:**
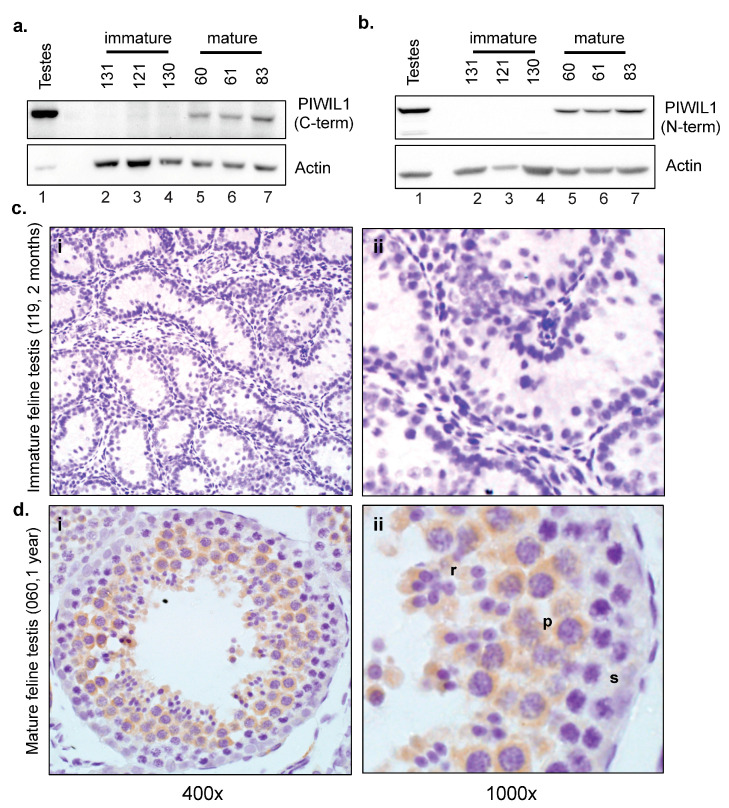
Expression of PIWIL1 protein is associated with level of sexual maturity (**a**) Western blot analysis of endogenous PIWIL1 expression in mature and immature feline testis tissues using C-terminal PIWIL1 antibody recognizing only PIWIL1-isoform 2. β-ACTIN used as a loading control. (**b**) Western blot analysis of endogenous PIWIL1 expression in mature and immature feline testis tissues using N-terminal PIWIL1 antibody able to recognize both PIWIL1 isoforms. β-ACTIN used as a loading control. (**c**) Photomicrographs of immunohistochemistry of PIWIL1 in immature feline testes. (**a**,**b**) Immature testis (2 months old); no staining was observed in pre-spermatogonia or spermatogonia; there was no specific staining in the tubules of the pre-pubertal kittens. Higher magnification shown in panel cii. (**d**) Photomicrographs of immunohistochemistry of PIWIL1 in mature feline testes. (**a**,**b**) Mature testis (1 year old); staining was seen in the cytoplasm of spermatocytes (*p*) but not spermatogonia (s). Staining was consistently observed in round spermatids (r), and in elongate spermatids when observed, with the exception of those with minimal cytoplasm. Higher magnification shown in panel dii. Formalin- and alcohol-fixed tissue. Original magnification 400×, increased magnifications at 1000×, no specific staining observed in matched negative controls, *n* = 3.

**Figure 6 ijms-24-09346-f006:**
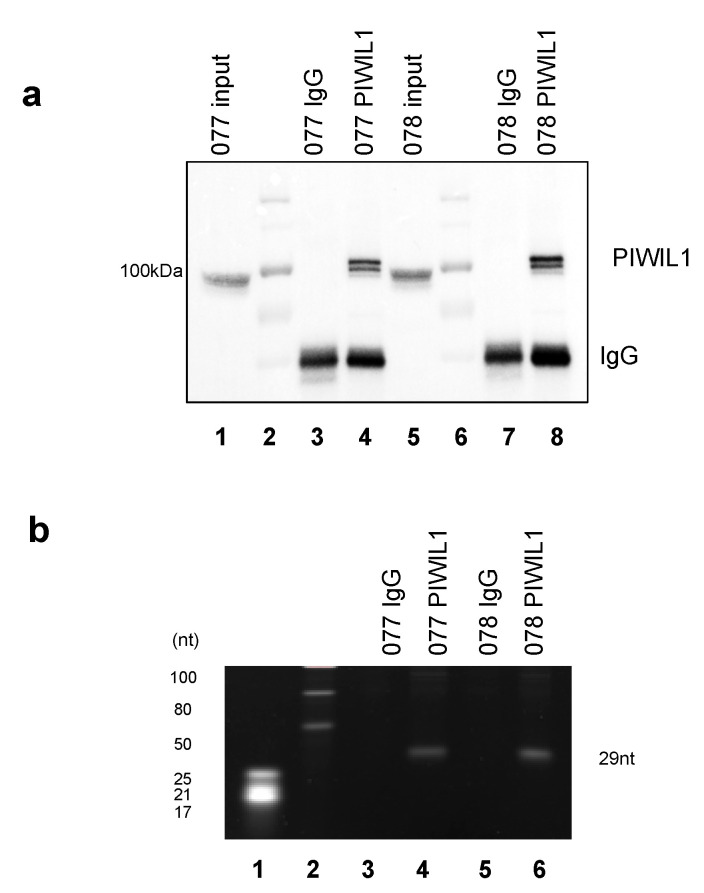
PIWIL1 interacts with a distinct 29 nt small RNA population. Feline mature testis lysates (*n* = 2) were immunoprecipitated with 3 µg of anti-PIWIL1-N-terminal antibody, able to recognize both PIWIL1-isoform 1 and PIWIL1-isoform 2. Eluents from both magnetic protein A beads alone (not shown), and magnetic protein A beads bound to 2 µg of rabbit IgG were used as negative controls. **Panel** (**a**) Representative Western blot of immunoprecipitated PIWIL1 samples. samples. PIWIL1-C terminal antibody was utilized to probe Western blots from anti-PIWIL1 (N-terminal) immunoprecipitated and control IgG immunoprecipitated testes samples as described in the methods. Lane 1: Non-precipitated input from sample 077, Lane 2: MWM, Lane 3: IP using control IgG with sample 077, Lane 4: IP of sample 077 using PIWIL1 N-terminal antibody, Lane 5: Input from sample 078, Lane 6: MWM, Lane 7: IP using control IgG with sample 078, Lane 8: IP of sample 078 using PIWIL1 N-terminal antibody. “Input” samples utilize 10% of the protein amount used in the IP experiments. Lower bands in Lanes 3, 4, 7, 8 represent secondary antibody binding to IgG from the immunoprecipitation reactions. **Panel** (**b**) RNAs bound to immunoprecipitated PIWIL1 were analyzed by denaturing TBE-UREA polyacrylamide gel electrophoresis and stained using SYBR gold. RNA extracted from immunoprecipitations performed with non-specific rabbit IgG was included as a negative control. Lane 1: low-range ssRNA ladder, Lane 2: microRNA marker, Lane 3: IP using control IgG with sample 077, Lane 4: IP of sample 077 using PIWIL1 N-terminal antibody, Lane 5: IP using control IgG with sample 078, Lane 6: IP of sample 078 using PIWIL1 N-terminal antibody. Band size quantified by gel analysis software using a log RF calculation based on the size of the marker nt. Representative gel shown.

## Data Availability

Not applicable.

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
