# Peer review of "cDNA Cloning of Feline PIWIL1 and Evaluation of Expression in the Testis of the Domestic Cat"

_ijms, 2023, doi:10.3390/ijms24119346_

Round 1

Reviewer 1 Report

The authors cloned PIWIL1 in cat and showed preliminary studies on it. However, most of the experiments need to be improved before publication.

Major points:

1.       The authors found cat PIWIL1 appears with sex maturity. But in mouse, PIWIL1 starts to express from ~14 dpp, an earlier stage before mouse mature. If true, this discrepancy is interesting. The authors should carefully investigate this with cat testis of different ages by both immunostaining and western blotting. In Fig 5d, the authors suggested PIWIL1 is expressed in spermatocytes. When the spermatocytes first appeared in cat testis? After mature?              

2.       Mammalian piRNAs size ranges from 26nt to ~32nt. Specific Piwi member protein binds specific piRNAs with specific length. For example, MIWI binds ~30nt pachytene piRNA, while MIWI2 and MILI favor shorter piRNAs. In Fig 6b, how can the authors tell the length of small RNAs pull downed by PIWIL1 as 29 nt? The authors should carefully identify the specific size of piRNA population associated with Feline PIWIL1 protein by 3’ end labelling and Urea Page, with anti- mouse PIWI member RNA IP as a positive control.

Minor points:

1.       Age information should be indicated for all immature and mature samples;

2.       The authors should number the lanes in all gels;

3.       RNA length measurement use “nt”, not “bp”;  

4.       There are numerous errors or redundant phrases present in current manuscript, should be carefully revised. Such as in legend of Figure 5, what does the “Higher magnification shown in panel b” mean? In legend of Figure 6, “L1, microRNA marker” appeared twice.

5.       The authors should discuss the potential reason under the discrepancy of PIWIL1 expression window between cat and mouse, if their finding is real.

Author Response

The authors cloned PIWIL1 in cat and showed preliminary studies on it. However, most of the experiments need to be improved before publication.

We thank the reviewer for their thoughtful consideration of our manuscript. The suggested changes have improved the quality of the study. We have made all corrections that were possible except for those we could not perform due to lack of available samples.

Major points:

  1. The authors found cat PIWIL1 appears with sex maturity. But in mouse, PIWIL1 starts to express from ~14 dpp, an earlier stage before mouse mature. If true, this discrepancy is interesting. The authors should carefully investigate this with cat testis of different ages by both immunostaining and western blotting. In Fig 5d, the authors suggested PIWIL1 is expressed in spermatocytes. When the spermatocytes first appeared in cat testis? After mature?            

The reviewer raises an important point with respect to the expression of PIWIL1 that we are, unfortunately, unable to address with available samples. Despite extensive efforts over the past several months, we have been unable to obtain additional testis samples – studies funded by the agencies that have supported these studies do not permit euthanasia of healthy animals, which would be necessary to obtain earlier samples. This was the primary reason for the delay in resubmission of this work. In order to partially address this question, we have modified the discussion (Lines 425 – 430) to describe the process more fully in cats and to incorporate the reviewer suggestion in minor point 5 below.

  1. Mammalian piRNAs size ranges from 26nt to ~32nt. Specific Piwi member protein binds specific piRNAs with specific length. For example, MIWI binds ~30nt pachytene piRNA, while MIWI2 and MILI favor shorter piRNAs. In Fig 6b, how can the authors tell the length of small RNAs pull downed by PIWIL1 as 29 nt? The authors should carefully identify the specific size of piRNA population associated with Feline PIWIL1 protein by 3’ end labelling and Urea Page, with anti- mouse PIWI member RNA IP as a positive control.

We agree completely with this comment. In our original manuscript we describe RNA-IP and urea/PAGE experiments which are essentially identical to those suggested by the reviewer (without end labelling). Our descriptions apparently did not make this clear, so we have modified the Results (Lines 284 - 310), and figure legend (Figure 6) in order to more accurately describe the studies and indicate how a size of 29nt was calculated for the piRNA population bound to immunoprecipitated PIWIL1.

Minor points:

  1. Age information should be indicated for all immature and mature samples;

Now included in the supplementary material (Table S4)

  1. The authors should number the lanes in all gels;

Lanes have been labelled as suggested

  1. RNA length measurement use “nt”, not “bp”;  

Length measurement description has been revised throughout the manuscript and in the figures.

  1. There are numerous errors or redundant phrases present in current manuscript, should be carefully revised. Such as in legend of Figure 5, what does the “Higher magnification shown in panel b” mean? In legend of Figure 6, “L1, microRNA marker” appeared twice.

Revisions have been made as suggested (see legends for Figures 5 and 6). Redundant phrases have also been removed.

  1. The authors should discuss the potential reason under the discrepancy of PIWIL1 expression window between cat and mouse, if their finding is real.

Discussion has been modified as described in response to point 1 (Lines 425 - 430).

Reviewer 2 Report

In these study author described that the PIWIL expression in mature feline testes.  

Major comment

1) Genome sequence aligment information could not be major results for      manuscripts. it can be supplementary results 

2) ‌In figure 4,  in addition, Figure 4b and 4d bot result are same. it is no different 

3) In figure 5, author described that PIWIL1 was found in cytoplasm of spermatocytes but not sertoli cells or spermatogonia  it is not clear with your staining result. Author need to co-staining with PIWIL1 and spermatocytes marker, or PIWIL1 and sertoli cell marker or spermatid marker 

(sertoli cell maker: SOX9, Vimentin etc..., Spermatocytes marker: SYCP3 etc..., spermatid marker: protamin, acrosin, PGK etc.. )

4) Accoridng to your previous sutdy " PIWIL1 is expressed in the canine testis ~~~ samll RNAs", PIWIL1 is expressed in immature canine testis. 

  could you describe why that is different between canine and feline testes. 

5) This study is not new. it is very similar with your  previous studies (" PIWIL1 is expressed in the canine testis ~~~ samll RNAs")  

 Author need to describe why this result is important and How can the results of this study be used in veterinary medicine or biology field? 

Minor comment 

1) Original image file does not contain the entire blot image (full shot). This is MDPI submission regulations 

Author Response

We thank the reviewer for their thoughtful consideration of our manuscript. The suggested changes have improved the quality of the study. We have made all corrections that were possible except for those we could not perform due to lack of available samples.

In these study author described that the PIWIL expression in mature feline testes.  

Major comment

1) Genome sequence aligment information could not be major results for   manuscripts. it can be supplementary results 

Table 1 has been moved to supplementary data.

2) ‌In figure 4, in addition, Figure 4b and 4d both results are same. it is no different 

We agree. Based on the reviewer’s suggestion, we have removed panels c and d from the manuscript to create a revised figure 4.

3) In figure 5, author described that PIWIL1 was found in cytoplasm of spermatocytes but not Sertoli cells or spermatogonia  it is not clear with your staining result. Author need to co-staining with PIWIL1 and spermatocytes marker, or PIWIL1 and sertoli cell marker or spermatid marker 

(sertoli cell maker: SOX9, Vimentin etc..., Spermatocytes marker: SYCP3 etc..., spermatid marker: protamin, acrosin, PGK etc.. )

Unfortunately, as described in the response to Reviewer 1, samples are no longer available for these additional experiments. We have altered the description in the results and figure legend to state that “PIWIL1 staining is observed in the cytoplasm of cells with the morphologic features of spermatocytes” and have removed the statements regarding Sertoli cells and spermatogoinia.

4) Accoridng to your previous sutdy " PIWIL1 is expressed in the canine testis ~~~ samll RNAs", PIWIL1 is expressed in immature canine testis. 

  could you describe why that is different between canine and feline testes. 

Description has been revised (Lines 415 - 416) to clarify the differences between the two species.

5) This study is not new. it is very similar with your  previous studies (" PIWIL1 is expressed in the canine testis ~~~ samll RNAs")  

 Author need to describe why this result is important and How can the results of this study be used in veterinary medicine or biology field? 

We have expanded the relevant section of the discussion (Lines 415 - 416) to address this issue

Minor comment 

1) Original image file does not contain the entire blot image (full shot). This is MDPI submission regulations 

 Full image blots have now been included in the supplementary material.

Round 2

Reviewer 2 Report

The manuscript has improved but author must change the image Figure 2 and 3. It is difficult to distinguish because image is stretched.